# Comparison of Immunogenicity and Safety of Diphtheria–Tetanus–Pertussis–Hepatitis B–*Haemophilus influenza* B (Bio Farma) with Pentabio^®^ Vaccine Primed with Recombinant Hepatitis B at Birth (Using Different Source of Hepatitis B) in Indonesian Infants

**DOI:** 10.3390/vaccines11030498

**Published:** 2023-02-21

**Authors:** Eddy Fadlyana, Kusnandi Rusmil, Meita Dhamayanti, Rodman Tarigan, Cissy B. Kartasasmita, Rini Mulia Sari, Muhammad Gilang Dwi Putra, Hadyana Sukandar

**Affiliations:** 1Child and Health Department, Faculty of Medicine, Padjajaran University, Hasan Sadikin Hospital West Java Indonesia, Bandung 40161, Indonesia; 2Surveillance & Clinical Research Division PT Bio Farma, Bandung 40161, Indonesia; 3Public Health Department, Faculty of Medicine, Padjajaran University, Bandung 40161, Indonesia

**Keywords:** diphtheria–tetanus–pertussis hepatitis B–*Haemophilus influenza* B vaccine, infants, protectivity, safety

## Abstract

Satisfying the needs of the national immunization program requires maintaining diphtheria–tetanus–pertussis (DTP)–hepatitis B (HB)–*Haemophilus influenza* B (Hib) production. Therefore, new hepatitis B sources are needed. This study aimed to evaluate the immunogenicity of the DTP–HB–Hib vaccine (Bio Farma) that used a different source of hepatitis B. A prospective randomized, double-blind, bridging study was conducted. Subjects were divided into two groups with different batch numbers. Healthy infants 6–11 weeks of age at enrollment were immunized with three doses of the DTP–HB–Hib vaccine after a birth dose of hepatitis B vaccine. Blood samples were obtained prior to vaccination and 28 days after the third dose. Adverse events were recorded until 28 days after each dose. Of the 220 subjects, 205 (93.2%) completed the study protocol. The proportion of infants with anti-diphtheria and anti-tetanus titers ≥ 0.01 IU/mL was 100%, with anti-HBsAg titers ≥ 10 mIU/mL was 100%, and with Polyribosylribitol Phosphate-Tetanus Conjugate (PRP-TT) titers > 0.15 µg/mL was 96.1%. The pertussis response rate was 84.9%. No serious adverse events related to the study vaccine occurred. The three-dose DTP–HB–Hib vaccine (Bio Farma) is immunogenic, well tolerated, and suitable to replace licensed-equivalent vaccines.

## 1. Introduction

A target of the sustainable development goals (SDGs) for 2030 is to decrease infant and child mortality [1]. The SDGs point to 3.3 states to eliminate Hepatitis B by 2030 [2]. The World Health Organization (WHO) has designed a strategy to eliminate hepatitis B by 2030. The Global Health Sector Strategy (GHSS) on viral hepatitis was adopted in 2016 at the World Health Assembly [3]. Hepatitis control programs were created for helping different countries by WHO. Based on the Polaris data, Indonesia is not on track to achieve its targets [4].

During the Coronavirus disease 2019 (COVID-19) pandemic, infant and child mortality rates were higher in lower- and middle-income countries (LMICs) [5,6]. More than one-fifth of the world’s children, especially those in LMICs, are still not fully vaccinated, and 22.4 million children are incompletely vaccinated at 12 months of age; these children are at risk of vaccine-preventable diseases that cause significant morbidity and mortality [7]. In Indonesia, the goals for infant and child mortality rates have not been met [8]. One of the reasons for the mortality of children under 5 years is delayed immunization [6,9].

Diphtheria has a worldwide distribution but is now uncommon in Western Europe and the USA; however, it is still endemic in many parts of the world, especially in Southeast Asia [10]. In 2017, a national outbreak infected 939 patients and caused 44 fatalities in 170 districts in 30 provinces in Indonesia [11]. Although active immunization programs have reduced the incidence of the disease, the case fatality rate stood at 3.5–12% in the 2000s [11,12]. The diphtheria outbreak revealed the inadequate vaccination coverage compared to one decade before [13].

Tetanus is a major cause of death in Asia and Africa, especially in rural and tropical areas where neonatal tetanus (infection of the newborn via perinatal infection of the cut umbilical cord) is common. Data from the WHO show that neonatal mortality from tetanus ranges from <5 to >60 per 1000 live births [14]. However, in Indonesia, the incidence of tetanus has decreased in recent years. Between 2000 and 2002, inpatients aged 15–44 years (43.34%) and outpatients aged >45 years (44.16%) had the highest incidence rates of tetanus in all hospitals in Indonesia; this may be attributed to the patient’s nature of work, including farming and carpentry. However, inadequate immunization is the ultimate cause [15].

Basic immunization in Indonesia was initiated on 1977 through the expanded program in immunization (EPI) [16]. The basic vaccines in Indonesia include bacilli Calmette-Guérin (BCG, against tuberculosis), diphtheria, tetanus, and pertussis (DTP), oral polio vaccine, measles, and hepatitis B (HB) [17]. In 2012, 48% of children in Indonesia did not complete the basic immunization regimen [18], and in 2018, only 57% of children had completed the basic immunization schedule [18]. This situation is caused by the access to the health care, education levels, information, religion and the knowledge of their parents [16,17,18]. Meanwhile, only 32% of the provinces in Indonesia have high vaccination coverage [18]. From 2020 to 2021 during the COVID-19 pandemic, the proportion of children completing the immunization schedule further decreased. In 2020, the immunization target was 92%, while coverage was only 84%; in 2021, these figures were 93% and 84%, respectively [19]. There are more than 1.7 million infants who did not receive basic immunization in 2019–2021 [13]. Research using worldwide data shows that 75% of zero doses of vaccine were from 14 countries and this includes Indonesia [16].

The DTwP–HB–*Haemophilus influenza* B (Hib) vaccine (Pentabio, Bio Farma, Bandung, Indonesia) combines the diphtheria and tetanus toxoids, inactive pertussis bacteria, HB surface antigen, and Hib, and underwent clinical trials in 2013 [16]. Today, the vaccine is safe for use as a booster [20]. Phase II clinical trials to determine the immunogenicity and safety of the Pentabio vaccine were conducted between July 2011 and January 2012. It was concluded that the Pentabio vaccine is safe and immunogenic and can be administered in healthy children [21]. Meanwhile, the Protectivity and Safety of DTP–HB–Hib (Pentabio) Vaccines in Infants, Batch Consistency, Multi Centers Trial, Phase III, and multicenter trial were conducted from August 2012 to January 2013. This study demonstrated that the Pentabio vaccine is immunogenic and well tolerated, and the results suggested that it may be used to support the childhood vaccination program in Indonesia [22,23,24,25]. A study about the safety profile of Pentabio following vaccination in Indonesian Infants was conducted in 2018. The most common systemic and local reactions were fever and pain, respectively; both persisted until day 1 after immunization. Pain intensity was mostly mild. No serious adverse events occurred during observation [22,23,24,25].

The true burden of HB, DTP, and Hib is multifactorial. Coordination with all of the stakeholders for the access and vaccine schedule is needed in future strategies. This study was conducted to evaluate the immunogenicity of the Pentabio vaccine, which uses a different source of HB.

## 2. Materials and Methods

### 2.1. Study Design and Population

This was a randomized, double-blind, bridging, prospective intervention study that was conducted at three primary health centers in Bandung city from October 2020 to October 2021. The subjects were randomized per treatment group. The randomization list was created with four blocks, using automatically generated randomization provided by the website www.randomization.com e.g., accessed on 13 October 2020. The subjects were divided into two groups based on the source of vaccine (Table 1).

The study population included healthy infants who were 0–3 days old at enrollment, were born between 37 and 42 weeks of gestation, weighed ≥ 2500 g at birth, with fathers, mothers, or legally acceptable representatives who were properly informed about the study and have signed the informed consent form, and with parents who agreed to comply with the indications of the investigator and with the schedule of the trial. Infants were excluded if they developed fever (axillary temperature ≥ 37.5 °C on day 0), had a history of an allergic reaction that was likely to be triggered by any vaccine component, if they had DTP, HB, or Hib infection, had a history of congenital or acquired immunodeficiency, had uncontrolled coagulopathy or blood disorders, had chronic illness, or immunosuppressive conditions, if they were undergoing immunosuppressive therapy, had received immunoglobulin therapy or blood products before starting or during the study, had received a vaccination other than oral polio and the BCG vaccine, were participating in another clinical study, or had a mother who was HbsAg or HIV positive (by rapid test).

Infants were withdrawn from the study if, after vaccination, they experienced acute illness, had an axillary temperature of 37.5 °C, or had received treatment likely to alter their immune response in the 4 weeks before vaccination (administration of intravenous immunoglobulin, systemic corticoids, and blood products). Additionally, infants were withdrawn if they were accidentally administered a non-investigational DTwP, HB, or Hib conjugate vaccine after the study vaccination, had encephalopathy and seizures within 7 days following the last injection, had an axillary temperature > 39.6 °C within 3 days following the last injection, inconsolable crying for more than 3 h following the last injection, had any clinically significant allergic reaction suspected to have occurred within 3 days following the last injection, had hypotonic hyporesponsive episodes within the 3 days following the last injection, and had thrombocytopenic purpura.

### 2.2. Vaccine

The recombinant Hepatitis B vaccine was an inactivated HbsAg produced in yeast cells (Hansenula polymorpha), using recombinant DNA technology. It was a whitish liquid that was produced by a yeast cell, genetically engineered in culture and carrying the relevant gene of the HbsAg. The Pentabio^®^ (Bio Farma) was a DTP–HB–Hib combination vaccine produced in forms of homogenous suspension containing the following: Diphtheria toxoid, tetanus toxoid, inactive Pertussis bacteria (whooping cough), non-infectious pure Hepatitis B surface antigen (HBsAg) and a non-infectious Hib component, which was a resulting conjugation of Haemophillus influenza type B and tetanus toxoid.

### 2.3. Study Vaccine

A randomized, prospective, double-blind, two-arm parallel group intervention study was conducted in Bandung city from October 2020 to October 2021. Subjects were divided into two groups: Group A received DTP–HB–Hib using a different source of hepatitis B, and Group B received DTP–HB–Hib (Pentabio^®^). After a birth dose of HB vaccine, healthy infants, 0–3 days of age at enrollment, were immunized with three doses of DTP–HB–Hib vaccine with an interval of 4 weeks. Blood samples were obtained before vaccination and 28 days after the third dose. Adverse events were recorded until 28 days after each dose.

### 2.4. Immunogenicity Assessment

Blood samples (4 mL) were collected in vacutainer tubes at visit 0 (V0) (cord blood at birth), V2 before DTP–HB–Hib vaccination, and V5 (28 days after the third dose of DTP–HB–Hib). After clotting at room temperature, blood samples were centrifuged at 3000 rpm for 15 min, and sera were aseptically separated into 2 aliquots (1.5 mL each) within 24 h after sampling. Each blood sample was labeled indicating the blood sampling stage (V0, V2, and V5), trial code, inclusion number, and subject’s initials. Sera were rapidly stored in a freezer at −20 °C/−80 °C. Temperature was monitored and documented on the appropriate form during the entire trial.

The anti-diphtheria, anti-tetanus, and anti-PRP-TT serological responses were tested by the ELISA method, and anti-pertussis was tested by the micro agglutination procedure. Anti-HBs ELISA was tested using a kit reagent from Abbott. Anti-hepatitis B titers were measured by an chemiluminescent microparticle immunoassay architect using a reagent kit on architect i 1000sr.

All tests (ELISA and microagglutination), except for anti-HBs, were conducted in the Immunology Laboratory of the Clinical Trial Department of Bio Farma. These tests were already validated by the Clinical Trial Department and approved by the Quality Assurance Division. This lab has been certified for ISO 14001:2004, ISO 9001:2008 and OHSAS 18001:2007. The test for anti-HBs was conducted in Commercial Laboratory, and had been assessed by Quality Assurance of Bio Farma and certificated for ISO 9001 and KAN (National Accreditation Committee) Jakarta, Indonesia. Any additional serological analysis on antigen(s) included in the vaccine tested in this trial may be performed if Bio Farma considers it necessary to further document the immunogenicity results of this trial or other trials.

### 2.5. Safety Assessment

The investigator assessed the intensity (code 1, 2, or 3), duration, and relation of each adverse event during the trial vaccines. Local and systemic reactions, expected or unexpected, occurring within 30 min, 72 h, and 28 days after each injection were evaluated by interviewing the parents on post-surveillance visits V1, V2, V3, and V4. The axillary temperature was measured for three days after each dose in the evening and/or at the time of the febrile peak, and the highest temperature (°C) was recorded in the diary card. The trial team recorded the information in the CRF. Any severe adverse event occurring within 28 days after each injection was also recorded.

### 2.6. Statistical Analysis

The sample size is determined based on a 95% confidence interval and the power of the test at 80%. Using the sample size formula for comparing two population proportions, the required sample size would be 110 in each group; this included the anticipation of a 10% dropout. 

The main evaluation criteria were defined as the percentage of infants with anti-diphtheria and anti-tetanus titers ≥ 0.01 IU/mL, anti-HBsAg titers ≥ 10 mIU/mL, and anti-PRP-TT titers > 0.15 μg/mL 28 days after the last injection of DTP–HB–Hib in Group A. The primary analysis was a comparison of the post-last dose seroprotection rates between groups A and B.

For immunogenicity analysis, the seroprotection and/or seroconversion rates at the last post-surveillance visit (V5) were presented as crude rates with 95% confidence intervals (CI) (computed using the exact binomial probability). Meanwhile, the seroprotection and/or seroconversion rates at V1 and V4 were presented as geometric means of titers (GMTs) with 95% CI.

For safety analysis, the number and percentage of subjects with at least one local or severe reaction (within 30 min, 72 h, and 28 days of each injection) and their frequency (global and broken down according to the size of the reaction, when appropriate) were assessed.

## 3. Results

### 3.1. Study Population

This study enrolled 220 newborns; 14 were dropped from the study due to withdrawn consent and the investigator’s decision, due to their health conditions. The subjects were divided into two groups based on whether they received the test or control vaccine. There were 109 subjects in Group A and 111 subjects in Group B. Study recruitment was started on 13 October 2020 and completed on 6 April 2021. The subjects were followed-up until 5 months of age; at this time, blood samples had been collected on V0, V2, and V5. All samples were evaluated from April to November 2021. The open blinding of the randomization code was conducted on 12 November 2021 and was attended by the sponsor and investigators. Code A represented the control product recombinant hepatitis B^®^ (Bio Farma) and Pentabio^®^, while code B represented the recombinant hepatitis B and DTP–HB–Hib (Bio Farma) vaccines using a different hepatitis B source (Figure 1).

The demographic characteristics of the subjects enrolled in each group are shown in Table 2. Of the 220 subjects enrolled in the study, Group A was comprised mostly of females (51.4%), whereas Group B was comprised mostly of males (57%). The mean age of the subjects was 0.53 ± 0.56 days.

The serological respones in the study are summarized in Table 3. Regarding immunogenicity, primary vaccination in Group B resulted in a protective level of 100%, 100%, 82.5%, 100%, and 96.1% for diphtheria, tetanus, pertussis, HB, and Hib, respectively.

The seroprotection and vaccine response rates for each antigen in the study are summarized in Table 4, and GMTs are listed in Table 5. The seroprotective antibody concentrations were not significantly different between the two groups. Consistent vaccine effects were demonstrated for all vaccine antigens. The upper limit of the 95% CI for the difference between the two groups in seroprotection or vaccine response rates was less than the predefined limit of 10% for all antigens.

### 3.2. Diphtheria

After completing the three-dose primary series, nearly all subjects in each group achieved a standard protective concentration of ≥0.01 IU/mL (97.1% in group A and 100% in group B) and ≥0.1 IU/mL (97.4% and 87.4%) (Table 4) against diphtheria. There was no significant difference in the GMC values (*p* = 0.270) and seroprotection rates for ≥0.01 and ≥0.1 IU/mL (*p* = 0.08 and *p* = 0.529) between the two groups. The percentage of infants whose antibodies increased by ≥4 times in each group was 63.7% and 62.1%, and the proportion of seronegative infants who became seropositive was 95.7% and 100%.

### 3.3. Tetanus

The tetanus standard protective antibody concentrations for ≥0.01 and ≥0.1 IU/mL were 100% for both concentrations and groups. There was no significant difference in the GMC values (*p* = 0.270) and seroprotection rates for ≥0.01 and ≥0.1 IU/mL (*p* = 1.00 and *p* = 1.00) between the two groups. The percentage of infants whose antibodies increased by ≥4 times in each group was 22.5% and 32.0%, and the proportion of seronegative infants who became seropositive was 100% for both.

### 3.4. Pertussis

As shown in Table 4, the proportion of patients who showed a response to pertussis antibodies in every criterion was 92.2%, 84.3%, 66.7%, and 30.4% for Group A, and 82.5%, 76.7%, 58.3%, and 35.0% for Group B. The GMT values were not significantly different between the groups. There was a significantly higher proportion of subjects with an anti-pertussis concentration of ≥40 (1/dil) (*p* = 0.038) after vaccination than before vaccination. The percentage of infants whose antibodies increased by ≥4 times in each group was 98% and 95.1%, and the proportion of seronegative infants who became seropositive was 94.5% and 92.6%.

### 3.5. Hepatitis B

Nearly all subjects in each group (99.0% and 100%) achieved seroprotective antibody concentrations ≥10 mIU/mL against the HB surface antigen after vaccination at birth and after the three-dose primary series (Table 4). The percentage of infants whose antibodies increased by ≥4 times was 89.3% and 97.1% in groups A and B, respectively; the difference between the groups was statistically significant (*p* = 0.027). The proportion of seronegative infants who became seropositive was 100%. The difference in the levels of anti-HBs GMTs was statistically significant at the V5 group (*p* = 0.001) (Table 5).

### 3.6. Hemophilus Influenzae Type b

After completing the three-dose primary series, ≥96% and ≥79% of subjects in each group had seroprotective anti-PRP concentrations of ≥0.15 μg/mL and ≥1.0 μg/mL, respectively (Table 4). The percentage of infants whose antibodies increased by ≥4 times in each group was >77%. The proportion of seronegative infants who became seropositive in each group was both 100%. Levels of anti-PRP GMTs were also comparable between the groups; a robust anti-PRP response was observed after the third dose in both groups (Table 4). There was no significant difference in the GMT values in both groups (*p* = 0.974).

### 3.7. Safety Assessment

Most of the patients did not develop adverse events within 30 min, 72 h, and 28 days after vaccination. Only redness and swelling within 72 h after the first dose of DPT–Hb–Hib was noted. No anaphylactic or severe reactions occurred within 30 min after any dose of the study vaccine.

### 3.8. Local and Systemic Reactions

Figure 2 and Figure 3 show the proportion of subjects in each group in whom local (injection site) and systemic reactions occurred after vaccination at birth and after the first, second, and third doses. Most local reactions in all vaccine groups were mild (Figure 2). The most frequently reported local reaction after each dose in each group was pain (>8%). Redness, swelling, and induration occurred with a similar frequency in both groups. Exploratory analyses showed that the incidence of redness and swelling after the first dose was significantly higher than the other doses. The most common solicited systemic reaction was irritability (Figure 3). Fever, which was mild or moderate in intensity and transient, occurred with a similar frequency between the two groups.

### 3.9. Serious Adverse Events

From the first dose to 35 days after the third dose, 26 serious adverse events occurred in 11 (10.6%) and 15 (14.5%) subjects in groups A and B, respectively. In 17 subjects, the serious adverse event was neonatal hyperbilirubinemia (Group A, *n* = 9; Group B, *n* = 8). Meanwhile, two subjects had ABO incompatibility and omphalitis. Five subjects (Group A, *n* = 2; Group B, *n* = 3) were hospitalized for 4–10 days due to bronchopneumonia. Two subjects with Hirschsprung disease, one subject with dengue hemorrhagic fever, and one subject with malnutrition were hospitalized; these conditions were considered as coincidental and unrelated to the study vaccine, as declared by the National Adverse Event Following Immunization committee.

## 4. Discussion

Vaccination is a public health strategy that effectively reduces the morbidity and mortality associated with various infectious diseases, especially in children [26]. The vaccination of children against diseases, such as diphtheria, tetanus, pertussis, poliomyelitis, HB, and invasive infections caused by Hib, prevents 2–3 million deaths and 750,000 children from being disabled each year worldwide [26,27,28,29]. There are multiple advantages to combining vaccines, including reducing the number of visits and injections, reducing patient discomfort, increasing compliance, and optimizing prevention. The World Health Organization (WHO) recommends that routine infant immunization programs include vaccination against Hib in the combined DTP–HB vaccine; this is because the combination vaccine has a similar efficacy in preventing these diseases to that of separate vaccines [30].

This study shows that the proportion of infants with anti-diphtheria and anti-tetanus titers ≥ 0.01 IU/mL was 100%, those with anti-HbsAg titers ≥ 10 mIU/mL was 100%, and those with anti-PRP-TT titers ≥0.15 μg/mL was 96.1% 28 days after the last dose. Meanwhile, the pertussis vaccine response rate was 82.5%. The results of this study are similar to those of previous reports [28].

In this study, the immunogenicity and seropositivity for diphtheria were good. In this study, transplacentally acquired antibodies for an anti-tetanus toxoid were present in almost all subjects (98.1%) before the primary vaccination series. Research in the United Kingdom reported significantly lower antibodies against tetanus toxoid antigens in infants in whom pre-immunization (maternally derived) antibody concentrations were already high [31]. Another study in India showed a significant rise in antibody titers for all components, except tetanus, one month after the last dose (administered at 6, 10, and 14 weeks of age) of a pentavalent DTwP–HB–Hib vaccine in Indian infants; this was attributed to the presence of maternal antibodies [32]. However, a meta-analysis reported that two-fold higher maternal antibody titers (pre-vaccination) were associated with lower post-vaccination antibodies against pertussis, tetanus, and other antigens [33].

Cases of newborns with high levels of transplacentally acquired anti-tetanus toxoid are common in Indonesia, where programs for the prevention of neonatal tetanus are implemented by vaccinating pregnant women. Neonatal tetanus can be prevented by vaccinating women of reproductive age with the tetanus toxoid either during or outside of pregnancy. For lifelong protection, the WHO recommends that an individual receives six doses (three primary plus three booster doses) of tetanus toxoid [34]. The three-dose primary series should be administered as early as 6 weeks of age, with subsequent doses administered within a minimum interval of 4 weeks between doses [34]. The three booster doses should preferably be given during the second year of life (12–23 months), at 4–7 years old, and at 9–15 years old. Ideally, there should be at least 4 years between booster doses [34].

Additionally, 61.2%, 80.6%, 17.3%, and 1.0% of subjects had seroprotective antibody concentrations against diphtheria, HB surface antigen, Hib, and pertussis before receiving the primary vaccination series. However, the low levels of anti-pertussis antibodies should be given attention. At the age of 2 months and after receiving HB immunization at birth, the HB vaccine only provided protection to 17.3% of the subjects; this increased to 100% after administering three doses containing HBsAg. Protection lasts for at least 20 years and is probably lifelong. The WHO does not recommend booster vaccinations for persons who have completed the three-dose vaccination schedule. However, the WHO recommends that all infants receive the HB vaccine as soon as possible after birth, preferably within 24 h, followed by two or three doses of HB vaccine at least 4 weeks apart to complete the vaccination series [30]. In areas with a high prevalence of HB infection, such as in Southeast Asia, perinatal transmission plays an important role in the high prevalence of the disease. In highly endemic areas, HB infection is most commonly transmitted either from mother to child at birth or from person to person in early childhood [30,31]. In countries where HB is endemic, early infant immunization is recommended. Since HB immunization coverage is much lower in Indonesia, combination with DTP is the best strategy to increase the HB immunization coverage [35].

Infants aged 2 months (6–10 weeks) have a high-risk of pertussis because only 1% have antibodies above the protection value; hence, these infants should receive immunization immediately. A previous study demonstrated that the pooled efficacy of the whole-cell pertussis (wP) vaccine against pertussis in children was 78%, but efficacy varied significantly among the vaccines [36]. A study from Australia demonstrated that the efficacy of the wP vaccine was highest among children aged 8–23 months [37]. Pertussis has a worldwide distribution and affects about 40 million people per year [38]. Before the introduction of effective vaccination programs, pertussis was common [39]. However, despite being vaccine-preventable, pertussis is still prevalent in Indonesia [40,41]; from 2016 to 2020, 274 clinical cases of pertussis were recorded [42]. The annual incidence of pertussis in Indonesia ranges from 0.0671–3.92 per 100,000 persons [40,41], which may be attributed to the limited availability of laboratory PCR testing as the gold standard for diagnosing pertussis [37,38,39].

Public health concerns before the implementation of Hib immunization programs included pneumonia and meningitis in young children caused by Hib [39]. There were approximately 8.13 million cases of severe infections in children under 5 years in 2000 [43,44,45]. In 2022, a study in Indonesia found that Hib is one of the pathogens responsible for causing pneumonia that requires hospitalization among Indonesian children aged 2–59 months old. Hib conjugate vaccine programs have alleviated the global burden of Hib meningitis in children [46].

The results of immunogenicity testing between the control and intervention groups showed similarly favorable results. Meanwhile, the GMT values of anti-HBs from the DTP–HB–Hib (Bio Farma) vaccine using a different hepatitis B source were significantly higher. A study of Pentabio in Iran showed similar results to this study [47].

The incidence of local and systemic reactions decreased with successive doses of primary vaccination [25]. This study found that pain and irritability were the most frequently reported local and systemic reactions, respectively, and fewer than 3% of local or systemic reactions were reported as severe after any dose in both groups. Fever of any severity was reported at low rates among all subjects after any dose. A previous study in Indonesia showed that the most common solicited local reaction was local pain, which mostly occurred within 30 min after any dose [25].

All vaccines were well tolerated; there were no differences in the rates of local and systemic reactions between the two groups. In the Pentabio group, 100% of the children reached protective levels of antibodies (seropositivity) against tetanus and diphtheria, 99% against HB, 98.1% against Hib, and 89.5% against pertussis. This shows that the Pentabio vaccine is safe and immunogenic and can be administered in healthy children [22]. A double-blind, randomized, batch consistency, prospective intervention, phase III, multi-center trial was conducted from August 2012 to January 2013. In total, 600 infants aged 6–11 weeks were distributed into two centers (Bandung and Jakarta). Each subject received three doses of the Pentabio vaccine in accordance with a randomization list and as their primary vaccination, with a 1-month interval between doses. Blood samples for a serology assessment were taken prior to and 28 days after the third dose. Safety was assessed immediately until 1 month after each injection through parent diary cards [23].

After primary vaccination with Pentabio, 99.3% of the subjects showed protective levels of antibodies against diphtheria, tetanus, HB, and Hib. The most common reactogenicity was local reactions, which were mild in severity. The immune responses and reactogenicity were consistent between the three different batch numbers of the Pentabio vaccine. This study demonstrates that the new pentavalent DTP–HB–Hib vaccine (Pentabio) is immunogenic and tolerable, and can be used to support a childhood vaccination program in Indonesia [23]. During the COVID-19 pandemic, this study was postponed to one year after we had planned, which is a limitation of this study.

In conclusion, our study shows that the three-dose DTP–HB–Hib (Bio Farma) regimen that uses a different source of hepatitis B is immunogenic, well tolerated, and suitable to replace licensed equivalent vaccines based on its immunological and safety profile. No vaccine-related serious adverse events or deaths were reported during the study.

## Figures and Tables

**Figure 1 vaccines-11-00498-f001:**
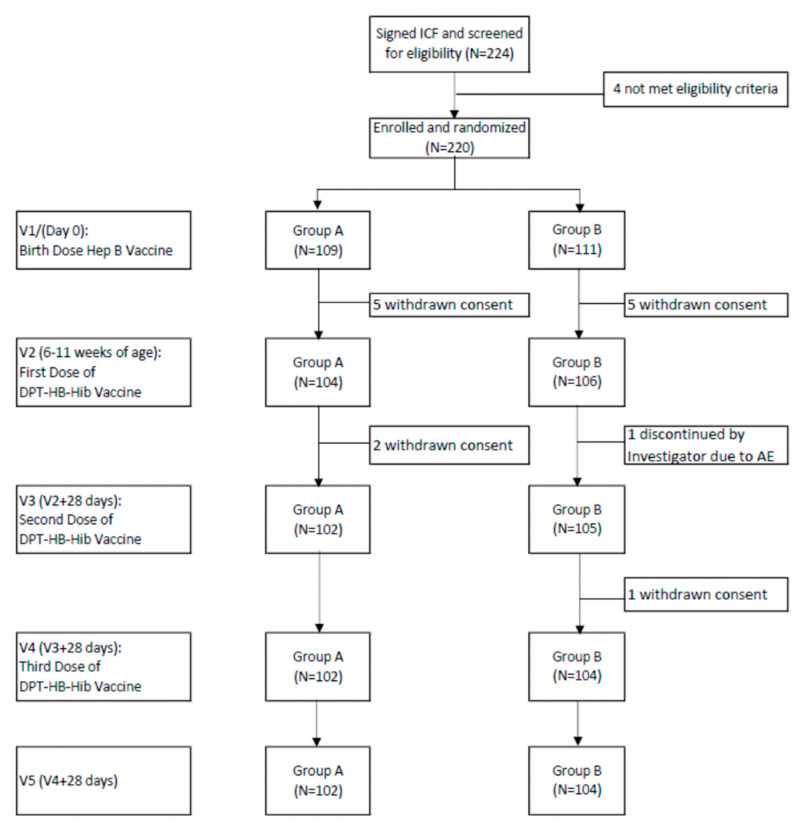
Participant’s flowchart.

**Figure 2 vaccines-11-00498-f002:**
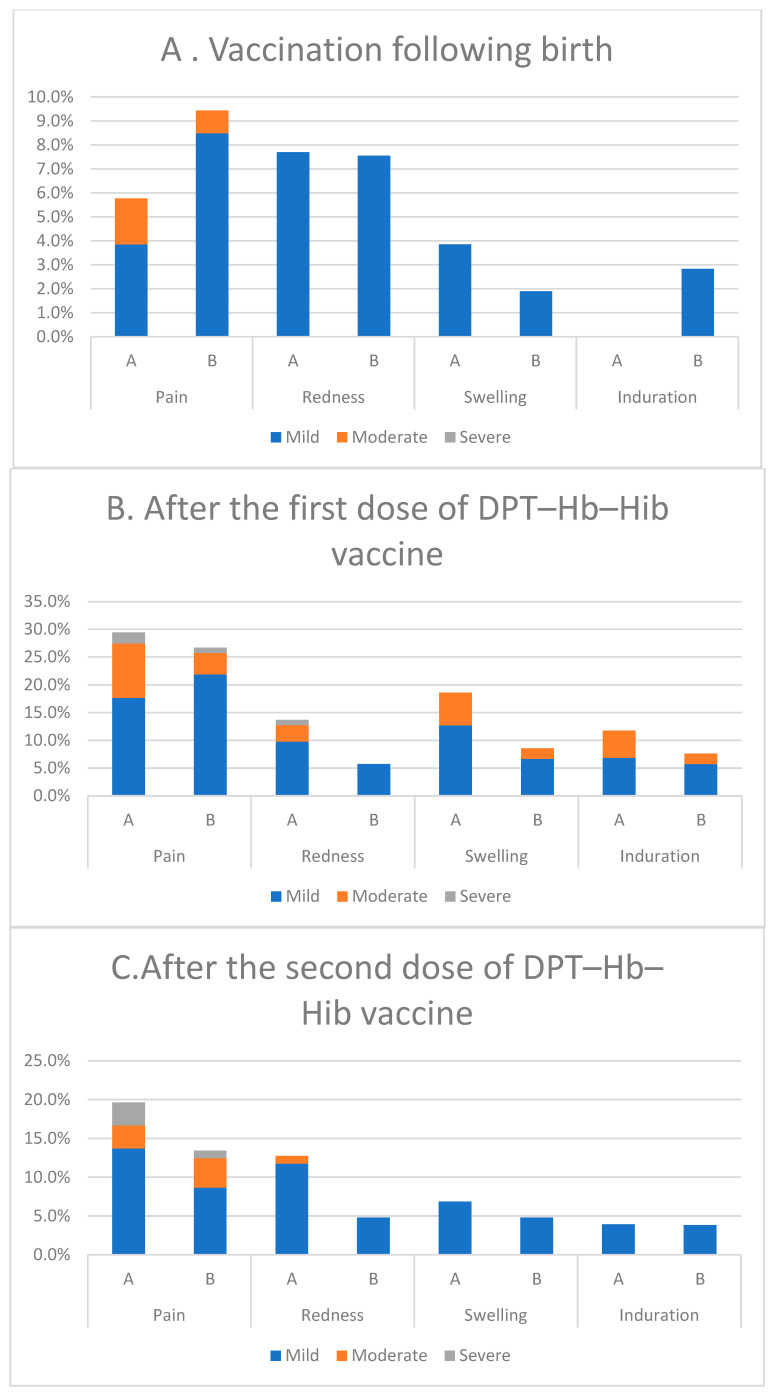
Local reaction.

**Figure 3 vaccines-11-00498-f003:**
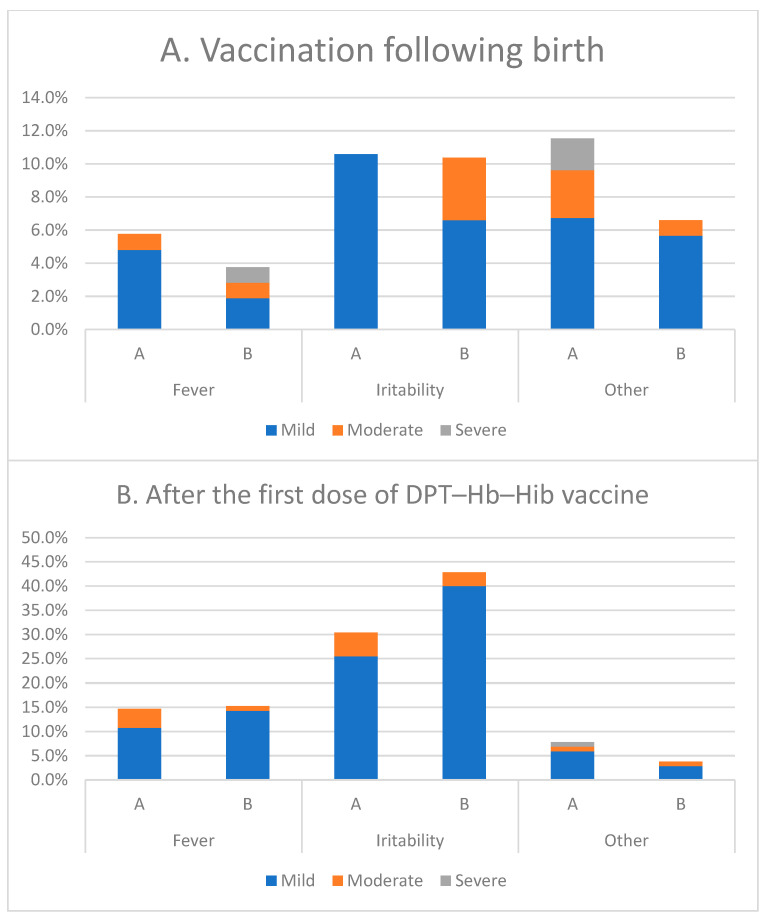
Systemic reaction.

**Table 1 vaccines-11-00498-t001:** Vaccines per intervention group.

Age	Group	0–3 (Days)	6–11 Weeks	10–15 Weeks	14–19 Weeks
0–3 days	A	Recombinant hepatitis B vaccine using different source	DTP–HB–Hib using different source of hepatitis B bulk	DTP–HB–Hib using different source of hepatitis B bulk	DTP–HB–Hib using different source of hepatitis B bulk
	B	Recombinant Hepatitis B^®^ vaccine (Bio Farma)	Pentabio^®^	Pentabio^®^	Pentabio^®^

**Table 2 vaccines-11-00498-t002:** Demographic characteristics.

Description	Control Product (A)	Investigational Product (B)	Total
N included	109	111	220
Gender Male *n* (%) Female *n* (%)	53 (48.6%) 56 (51.4%)	57 (51.4%) 54 (48.6%)	110 110
Age (days)			
Mean ± SD *	0.51 ± 0.55	0.54 ± 0.57	0.53 ± 0.56
Min; Max	0; 3	0; 2	0; 3

* SD: standard deviation.

**Table 3 vaccines-11-00498-t003:** Serological responses to diphtheria, tetanus toxoid, hepatitis B, and *Haemophilus influenza* B antigens in Group B after immunization.

Antibody Response	Pre-Vaccination/V2 (*n* = 103)	Post-Vaccination/V5 (*n* = 103)
Anti-D ≥ 0.01 IU/mL *n* (%)	63 (61.2)	103 (100)
Anti-T ≥ 0.01 IU/mL *n* (%)	101 (98.1)	103 (100)
Anti-PRP-T ≥ 0.15 μg/mL *n* (%)	83 (80.6)	99 (96.1)
Anti-pertussis ≥ 40 1/dil *n* (%)	1 (1.0)	85 (82.5)
	**Pre-vaccination/V0** **(** * **n** * ** = 104)**	**Post-vaccination/V5** **(** * **n** * **= 103)**
Anti-HBsAg ≥ 10 mIU/mL *n* (%)	18 (17.3)	103 (100)

**Table 4 vaccines-11-00498-t004:** Summary of seroprotective antibody rates.

Antibody	Assessment	Criterion	Group A			Group B			*p*-Value
			N	%NSP	95% CI	N	%NSP	95% CI	
Diphtheria	Pre-vaccination/V2	≥0.01 IU/mL	55	53.9	44.3–63.3	63	61.2	51.5–70.0	0.294
		≥0.1 IU/mL	31	30.4	22.3–39.9	37	35.9	27.3–45.5	0.400
	Post-vaccination/V5	≥0.01 IU/mL	99	97.1	91.7–99.0	103	100	96.4–100	0.080
		≥0.1 IU/mL	90	97.4	79.6–92.5	90	87.4	79.6–92.5	0.529
Tetanus	Pre-vaccination/V2	≥0.01 IU/mL	100	98	93.1–99.5	101	98.1	93.2–99.5	1.000
		≥0.01 IU/mL	93	91.2	84.1–95.3	99	96.1	90.4–98.4	0.147
	Post-vaccination/V5	≥0.1 IU/mL	102	100	96.4–100	103	100	96.4–100	1.000
		≥0.01 IU/mL	102	100	96.4–100	103	100	96.4–100	1.000
Pertussis	Pre-vaccination/V2	>40 1/dil	4	3.9	1.5–9.6	1	1.0	0.17–5.3	0.212
		>80 1/dil	1	1.0	0.17–5.35	1	1.0	0.17–5.3	1.000
		>160 1/dil	0	0	0–3.63	1	1.0	0.17–5.3	1.000
		>320 1/dil	0	0	–	0	0	–	-
	Post-vaccination/V5	>40 1/dil	94	92.2	85.3–96.0	85	82.5	74.1–88.6	0.038 *
		>80 1/dil	86	84.3	76.0–90.1	79	76.7	67.7–83.3	0.169
		>160 1/dil	68	66.7	57.1–75.1	60	58.3	48.6–67.3	0.214
		>320 1/dil	31	30.4	22.5–33.9	36	35.0	26.4–44.6	0.487
Hepatitis B	Pre-vaccination/V0	≥10 mIU/mL	19	18.1	11.9–26.5	11	10.5	5.9–17.8	0.115
	Pre-vaccination/V2	≥10 mIU/mL	27	26.0	18.5–35.1	18	17.3	11.2–25.7	0.130
	Post-vaccination/V5	≥10 mIU/mL	101	99.0	94.6–99.8	103	100	96.4–100	0.498
PRP (Hib)	Pre-vaccination/V2	≥0.15 µg/mL	77	75.7	66.3–82.8	83	80.6	71.9–87.1	0.378
		≥1.0 µg/mL	5	4.9	2.1–11.0	4	3.9	1.5–9.6	0.748
	Post-vaccination/V5	≥0.15 µg/mL	101	99.0	94.6–99.8	99	96.1	90.4–98.5	0.369
		≥1.0 µg/mL	81	79.4	70.6–86.1	84	81.6	73.0–87.9	0.699

* *p*-value was calculated using the Chi-square test/Fisher’s exact test.

**Table 5 vaccines-11-00498-t005:** Summary of geometric mean antibody titers (per-protocol immunogenicity population).

Antibody	Assessment	Group A		Group B		*p*-Value
		GMT, IU/mL	95% CI	GMT, IU/mL	95% CI	
Diphtheria	Pre-vaccination/V2	0.1465	0.0088–0.2431	0.0201	0.0122–0.03323	0.440
	Post-vaccination/V5	0.3260	0.2419–0.4393	0.4215	0.3380–0.5256	0.270
Tetanus	Pre-vaccination/V2	0.5950	0.4557–0.7770	0.6414	0.4944–0.8320	0.678
	Post-vaccination/V5	1.0268	0.8929–1.1808	1.2653	1.0691–1.4976	0.065
Pertussis	Pre-vaccination/V2	5.8462	5.2944–6.4570	5.606	5.128–6.117	0.596
	Post-vaccination/V5	133.179	106.346–166.783	105.419	80.868–137.423	0.407
Hepatitis B	Pre-vaccination/V0	0.3347	0.1551–0.72	0.1986	0.100–0.394	0.491
	Pre-vaccination/V2	1.4824	0.823–2.669	0.5458	0.280–1.063	0.076
	Post-vaccination/V5	533.303	415.090–85.184	2372.378	1905.03–2956.87	0.001 *
PRP-Hib	Pre-vaccination/V2	0.1643	0.1156–0.2336	0.2075	0.1529–0.2816	0.252
	Post-vaccination/V5	3.4965	2.6706–4.7936	2.9457	2.0478–4.2372	0.974

* *p*-value was calculated using the Mann–Whitney test.

## Data Availability

Data availability will be available on the main site of study. Contact the author for future access.

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
