# Peer review of "Comparison of Immunogenicity and Safety of Diphtheria–Tetanus–Pertussis–Hepatitis B–Haemophilus influenza B (Bio Farma) with Pentabio® Vaccine Primed with Recombinant Hepatitis B at Birth (Using Different Source of Hepatitis B) in Indonesian Infants"

_vaccines, 2023, doi:10.3390/vaccines11030498_

Round 1

Reviewer 1 Report

Dear Editor,

The manuscript titled “Comparison of Immunogenicity and Safety of diphtheria–tetanus–pertussis (DTP)–hepatitis B (HB)–Haemophilus influenza B (Hib)(Bio Farma) with Pentabio® vaccine Primed with Recombinant Hepatitis B at Birth (using different source of Hepatitis B) in Indonesian Infants” by Fadlyana et al., is an interesting paper aiming to evaluate the immunogenicity of the DTP–HB–Hib vaccine (Bio Farma) that used a different source of hepatitis B antigen. A prospective randomized, double-blinded, bridging study was conducted. The authors concluded that the three-dose DTP–HB–Hib vaccine (Bio Farma) is immunogenic, well tolerated, and suitable to replace licensed-equivalent vaccines.

The manuscript is well written and methodologically correct. Minor English spell check is required. Some issues need to be addressed before publications.

Minor revisions:

·      In the abstract and main text, the authors mention a “different source of hepatitis B”, referring to the composition of the experimental vaccine. The description of vaccine composition and the nature of this new “hepatitis B sources” should be described in more details.

·      Abstract: the PRP-TT acronym should be clarified when used for the first time.

·      Line 47: … and hepatitis B (HB) [7].

·      In table 1 groups are referred to as group 1 and 2, while in the text and following tables and figures they are named A and B. Please be consistent with group definition. 

·      Line 127: PRP-TT, see above

·      Line 163: V0 is not defined. In figure 1 day 0 corresponds to V1. Please clarify this point in the main text and figure 1.

·      Line 199: Percentages refer to group A and B, respectively?

·      Line 212: “40 1/dil” as a measure unit needs to be further clarified. 

·      Lines 222-223: GMT shoud be reported in the text, to clarify in which group the titer was significantly hgier. 

·      Line 405-406: the sentence should be rephrased to improve clarity.

Author Response

Dear Reviewer, 

Thank you for your input, i have been adding the clarification on the word file and hope will be useful.

Thanks

Eddy

Reviewer 2 Report

The scientific quality of the paper is very strong and is addressing very important issue. Paper should be published on priority. Important scientific data is reported in the paper. I have few minor comments.

Title of the paper is very long, please reduce the length of title.

Abstract is well written. There is a heading of “objectives” in the abstract. please remove it.

Increase the length of introduction

Introduction paragraph 2: High number of Indonesian children are missing vaccinations. What is reason? vaccine coverage is low or there are issues of vaccine refusal, please write.

There is a Global Burden of Disease paper by University of Washington on childhood coverage in 204 countries published in The Lancet (398: 10299, page 503-521, 2021). Please add latest data on global and local vaccine coverage from the paper.

https://www.sciencedirect.com/science/article/pii/S0140673621009843

WHO has designed a strategy to eliminate hepatitis by 2030. Birth dose hbv vaccination and hbv vaccination in children, both are included in the strategy. Add in your paper about the WHO hepatitis elimination targets, how much progress is made and discuss how inclusion of hbv will help in achieving SDGs and hepatitis elimination. Suggested papers are data from Polaris USA and GBD University of Washington and other groups.

https://www.sciencedirect.com/science/article/pii/S2468125322001248

https://www.sciencedirect.com/science/article/abs/pii/S2468125318300566

https://www.ncbi.nlm.nih.gov/pmc/articles/PMC6262254/

Author Response

Dear Reviewer, 

Thank you for your input, i have been adding the clarification on the word file and hope will be useful.

Thanks

Reviewer 3 Report

Reviewer’s comments: Comparison of Immunogenicity and Safety of diphtheria–tetanus–pertussis (DTP)–hepatitis B (HB)–Haemophilus influenza B (Hib)(Bio Farma) with Pentabio® vaccine Primed with Recombinant Hepatitis B at Birth (using a different source of Hepatitis B) in Indonesian Infants. 

In this study, the authors compared the immunogenicity and safety of two pentavalent vaccines administered to infants at 6-11 weeks, 10-15 weeks, and 14-19 weeks of age. The differences between the two vaccines were the source of HBsAg used in the vaccine. 

Comments 

  1. In the last paragraph of the introduction part, the authors should clarify the reasons why a different source of HepB surface antigen has to come into play in vaccine manufacturing (i.e. the reduced supply of the original source, the accessibility or feasibility of the new source, etc.). Besides, there should be more details regarding the source of the HepB surface antigen of the original PentaBio vaccine. 
  2. Line 70: Please add the full name of HP 
  3. Methods section: In section 2.2, the authors should provide information on the component and amount of antigen used in the study and control vaccines. 
  4. Methods section: In section 2.3 (Immunogenicity assessment), the authors should provide more details or references on the anti-pertussis testing by the microagglutination methods Besides, the author should provide the reference on the cut-off for seropositivity against pertussis. 
  5. In the statistical analysis, sample size calculation should be explained. 
  6. Line 148-151: Should it be V5 instead of V4 to demonstrate the seroprotection and seroconversion rate? 
  7. In the results section (safety assessment), the authors should add a Figure title and Figure legends and label every Figure with A, B, C, and D to guide the readers when going through the Figures. For the systemic reactions, please define what symptoms represent “others.” 
  8. Two paragraphs between Lines 297-313 should be removed from the discussion section. These could be moved to the introduction part. 
  9. In the discussion section, the authors should discuss more on the comparison of the immunogenicity of Pentanbio to other WHO’s approved childhood combined vaccines.  
  10. The authors should consider naming the BioFarma regimen using a different source of hepatitis B in a shorter version (i.e., Biofarma-newHepB or Biofarma-rHepB). It will be easier for the reader to follow this study in which the immunogenicity and safety of Pentabio® and Biofarma-rHepB were compared.

Author Response

(The authors gave the same response as above.)
